# Origami-based integration of robots that sense, decide, and respond

Wenzhong Yan [1] ✉, Shuguang Li [2,3], Mauricio Deguchi [1], Zhaoliang Zheng[4], Daniela Rus [2] & Ankur Mehta[4]

Origami-inspired engineering has enabled intelligent materials and structures to process and react to environmental stimuli. However, it is challenging to achieve complete sense-decide-act loops in origami materials for autonomous interaction with environments, mainly due to the lack of information processing units that can interface with sensing and actuation. Here, we introduce an integrated origami-based process to create autonomous robots by embedding sensing, computing, and actuating in compliant, conductive materials. By combining flexible bistable mechanisms and conductive thermal artificial muscles, we realize origami multiplexed switches and configure them to generate digital logic gates, memory bits, and thus integrated autonomous origami robots. We demonstrate with a flytrap-inspired robot that captures 'living prey', an untethered crawler that avoids obstacles, and a wheeled vehicle that locomotes with reprogrammable trajectories. Our method provides routes to achieve autonomy for origami robots through tight functional integration in compliant, conductive materials.

Origami robots are autonomous machines created through a folding-based process borrowed from the ancient art of origami[1]—folding patterned two-dimensional (2D) sheets or even linear threads to complex three-dimensional (3D) objects. This folding-based strategy has endowed origami robots with advantages over conventional rigid and bulky robots, including rapid prototyping[2,3], high specific strength[4,5], built-in compliance for safe interaction with humans[6], low cost and high accessibility[7,8], and etc.[9–16]. Yet, almost all origami robots still rely on rigid semiconductor-based electronics and auxiliary transducers for sensing, control, and actuation to interact autonomously with their environments[1].

This dependency on rigid semiconductor-based electronics often restricts the potential of origami robots. Equipping external semiconductor-based electronics requires system integration thus increasing the complexity and weight of the resulting robots. These disadvantages mainly result from the undesired information transmission in the electro-mechanical interface[17]. The mismatch of stiffness between rigid electronics and the compliant bodies increases the

difficulty of design, fabrication, and assembly[18]. Semiconductor-based electronics are typically vulnerable to adversarial environmental events, e.g., radiation and physical impact, which limit their applications[19]. The logistic needs on-site could restrict robotic rescuers involved in disaster reliefs and first aid in resource-constrained locations. The dependency on semiconductor-based electronics might inhibit the promised accessibility of the folding-based method[20]. Therefore, it is desirable to develop an alternative method for origami robots to achieve autonomy by embedding sensing, computing, and actuation into compliant materials[21]. This may lead to a new class of origami robots, with levels of autonomy similar to their rigid semiconductor-based counterparts, while maintaining the favorable attributes associated with origami folding-based fabrication[1,17,22].

There have been increasing efforts in investigating the feasibility of integrating smart materials into origami structures and mechanisms to realize desired functionalities, including sensing, computing, communication, and actuation[23–26]. This parallels a broader exploration into non-traditional approaches to achieve information processing

[1]Mechanical and Aerospace Engineering Department, UCLA, Los Angeles, CA, USA. [2]Computer Science and Artificial Intelligence Laboratory, MIT, Cambridge, USA. [3]Department of Mechanical Engineering, Tsinghua University, Beijing, P.R. China. [4]Electrical and Computer Engineering Department, UCLA, Los Angeles, CA, USA. ✉e-mail: wzyan24@g.ucla.edu

and control across a range of disciplines; this has led to the opportunity of using mechanical computing systems to augment traditional electronic computing systems[27] in various fields, including soft robotics[28–33], microfluidics[34,35], mechanics[19,36,37], and beyond[38]. To autonomously interact with the environment through integrating smart origami materials, an analogical sense-decide-act loop that emulates the language and structure of conventional semiconductor-based architecture should be formulated. This requires computing units that process information[39], sensors that receive signals from the environment[40], and actuators that execute commands to implement the response upon the feedback[41]. Furthermore, those three classes of components must form an ecosystem that accommodates both signal transmission and energy transduction. A few components and some of their assemblies have been demonstrated individually[17,18,25,42,43]. However, it is still very challenging to build integrated autonomous origami robotic systems mainly due to the lack of suitable computing elements that can interface with available sensing and actuating components[42]. High resistance or energy loss of building components[44] and complicated fabrication[39] of current computing architectures also contribute in part to the challenge. To the best of our knowledge, origami robots have not been demonstrated that can autonomously interact with the environment with sensing, computing, and actuating capabilities fully embedded in compliant materials.

Here, we report an integrated process to create autonomous origami robots using functional compliant, conductive materials (see Supplementary Fig. 10 for the overall workflow). Figure 1a illustrates the concept of this process, in which robots can be constructed entirely from sheet and thread materials via cut-and-fold processing based on the fundamental enabling mechanism, the origami multiplexed switch (OMS, Fig. 1b). The resulting robots are referred to as OrigaMechs, short for Origami MechanoBots. We have demonstrated the integration of components critical to this vision in Fig. 1c, d, and e

(see Supplementary Table 3 for the design space). The OMS is created by incorporating bistable beams and conductive resistive actuators–conductive super-coiled polymer (CSCP) actuators[45]. Based on the OMS, origami logic gates (i.e., NOT, AND, OR gates with functional completeness) are devised; combinational logic and circuits, namely NAND and NOR gates, are also developed to show the capability of our system for cascading and more sophisticated computation. Therefore, the sense-decide-act loops necessary for autonomous interactions of robots can be built around origami logic with widely available electrically-mediated sensing and actuating mechanisms[36,46]. We demonstrate our method in three integrated autonomous origami robots that can sense, decide, and react to environmental stimuli. This method opens up new design space for autonomous origami machines that are less systematically complex, low cost, lightweight, and robust to adversarial environmental factors (e.g., magnetic field, radio frequency signal, electrostatic discharge, and mechanical deformation). Our work represents an essential step towards the realization of highly integrated and robust untethered origami robots, as well as intelligent machines.

## Results

### Design of the origami multiplexed switch

An OMS acts as a 2-to-1 multiplexer that selects between two analog or digital input signals ($V_+$ and $V_-$) and forwards the chosen input based on a selection signal, $V_S$ (and its complementary $V_R$, see Fig. 1b). The logic function of the OMS can be expressed as $Q = V_R \cdot V_- + V_S \cdot V_+$. The OMS mainly consists of one bistable beam and two CSCP actuators. One end of each actuator is attached to the bistable beam while the other is fixed on an origami framing. Thus, the actuators can drive the bistable beam switch between two stable states to control the on/off states of two electrical poles on the beam. When the beam is pulled by the left CSCP actuator, it snaps to the left stable state with the bottom

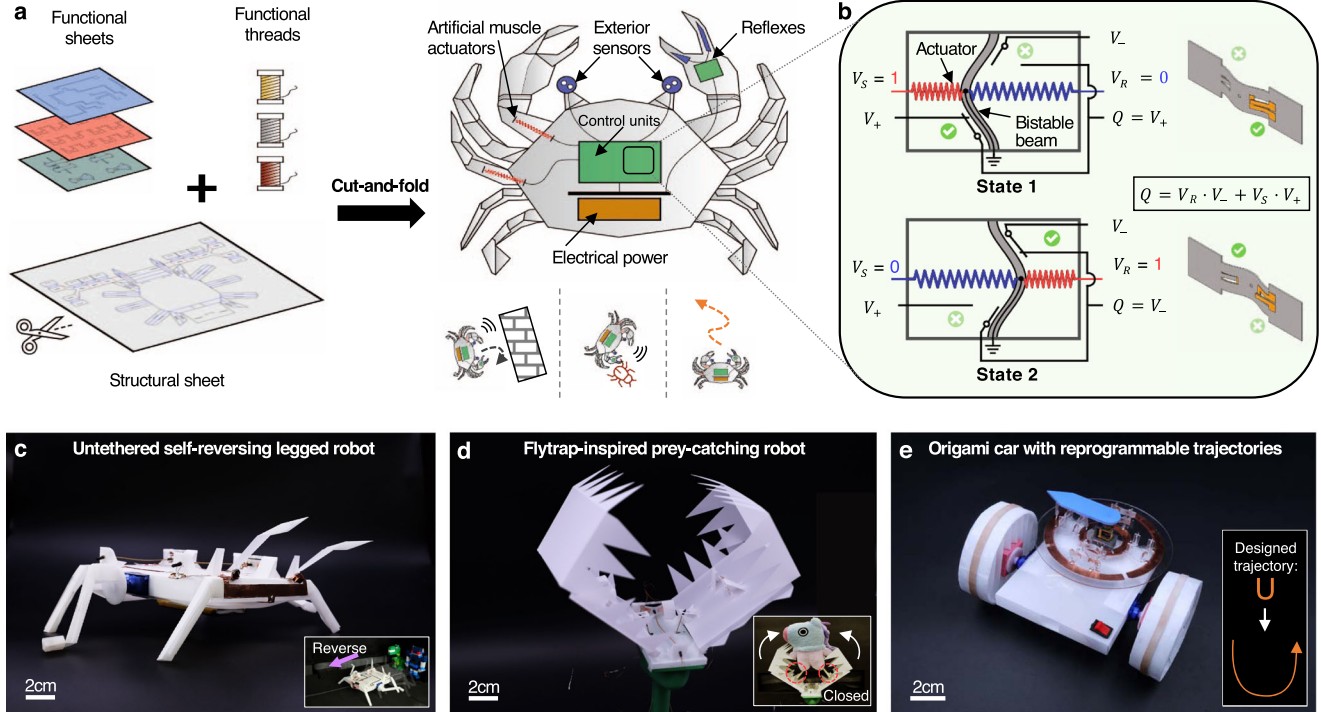

**Fig. 1 | Autonomous robots with sensing, computing, and actuating tightly integrated in compliant origami materials. a** A concept for an autonomous robot entirely created from functional sheet materials and threads through cut-and-fold processing. **b** The fundamental building unit of the autonomous robot is the origami multiplexed switches (OMS). The OMS can select between two input signals (i.e., $V_+$ and $V_-$) and forward the selected one on-demand according to the selection signal $V_S$ (with its complementary $V_R$). **c** An OMS-based crawler that can autonomously detect obstacles and execute decision-making to reverse its crawling direction. **d** A flytrap-inspired robot that can distinguish active objects from static stimuli and close its leaves to capture its 'prey'. **e** An untethered car that can locomote along trajectories programmed in OMS-based origami memory.

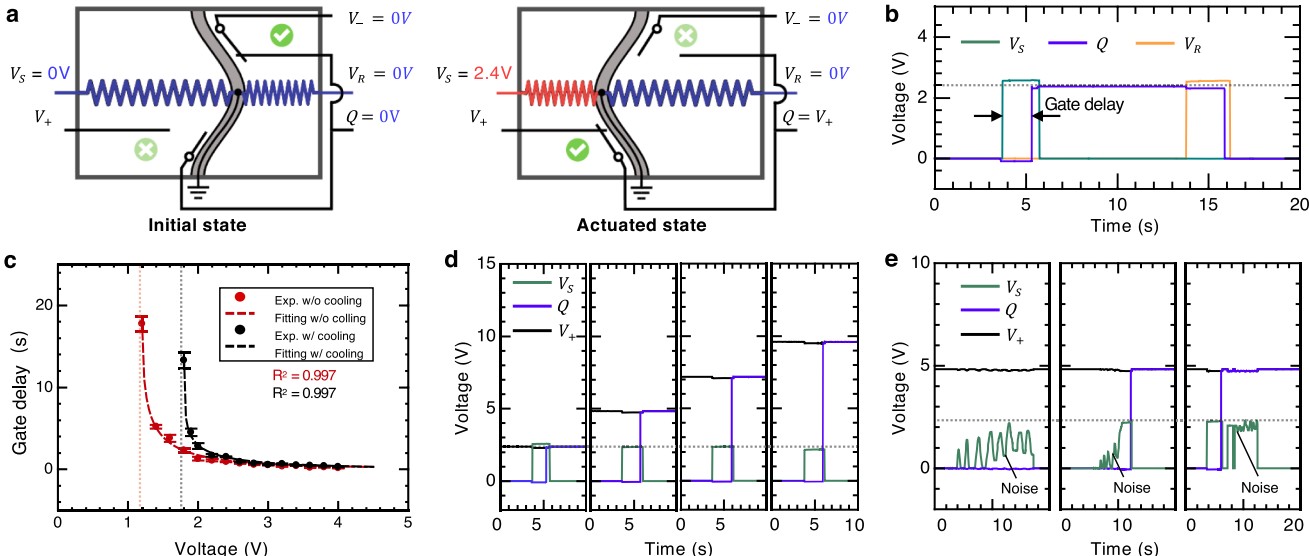

**Fig. 2 | Characterization of OMS. a** Two states of the OMS with labels of two inputs, $V_+$ and $V_-$, two control signals ($V_S$ and $V_R$), and one output, $Q$. **b** Typical operation of the OMS with $V_S = 2.4$ V, showing a gate delay of -1.5 s. Before the next cycle of operation, a voltage, $V_R$ is applied to switch the bistable beam back to reset the OMS. **c** Gate delay of OMS as a function of selection voltage, $V_S$, with/without cooling air. Error bars represent standard deviations obtained from three separate experiments. **d** The output of the OMS for different supply voltages of $V_S$, functioning as a relay. **e** Response of the OMS to three pulses as control inputs, indicating its robustness to noise for binary operation. Random noise is superposed to all pulses at different phases.

pole closed and the top pole open, leading to an output $Q = V_+$. Otherwise, the output changes to $V_-$. The mechanism of the bistable beam is described in Supplementary Fig. 1. The snap-through instability endows the switch with two properties. First, the state of the switch is binary ('open' or 'closed'), which allows unambiguous control, regardless of the uncertainties resulting from the nonlinear contracting of the actuators. This feature also enables it to function as a filtered touch sensor when the bistable beam is exposed to external mechanical stimuli[47]. Second, the switch requires power only when it changes between the two states; otherwise, it remains in the previous state, extensively reducing energy consumption[19]. The state of the switch is controlled by a pair of CSCP actuators[45]; the switch changes its states when the corresponding actuator drives the bistable beam reaching its switching point. The CSCP actuator acts as a thermal actuator and can be driven through Joule heating (similar to shape memory alloy actuators[48]).

The design of the tabs (forming the poles) on the bistable beam is critical. The two tabs naturally rest as cantilever beams. When the beam is at either state, the beam compresses the corresponding tab while releasing the other. Thus, the compressed tab forces the corresponding pole consisting of two copper tape terminals closed. The length and the angle of the tabs can be chosen such that the contact starts and finishes within the snapping motion of the beam. The width of the tab should be configured to be capable of applying enough pressure on the pole to reduce contact resistance. Our optimized design of the tab has a width of 1.2 mm, a length of 4.0 mm, and a folded angle of -30°. The folding line of the tab is aligned with the boundary of the bistable beam (with its geometry parameters listed in Supplementary Table 1). For the current design, we have an average ON resistance of 0.6 ohms and an OFF resistance of -1.2 megohms, resulting in a particularly high ON/OFF ratio (-10⁶). Also, the ON resistance is sufficiently low that it is suitable for the control of a wide range of robotic systems, which we demonstrate by using OMS or integrated devices to drive high-current CSCP actuators with low input voltage (<3 V). This low-resistance nature promises that the output signal of a logic unit can drive a large number of inputs of downstream logic gates without additional interfacing circuitry; this large fan-out endows our proposed architecture with the capability to build

complex circuits and systems. Coupling the instability of the bistable beam and tab design leads to binary, opposite states of two poles ('open/closed') on the bistable beam, with hysteretic switching behavior (Supplementary Fig. 22).

## Characterization of the origami multiplexed switch

Figure 2a shows an OMS that acts as a multiplexer between two different sources of electrical power ($V_+$ and $V_-$). $V_-$ is grounded while $V_+$ is connected to a constant positive supply voltage. Initially, the beam rests at the right stable state, the output, $Q$, of the switch is 0 V. When a control voltage $V_S$ (=2.4 V) is applied, the beam snaps left, changing its output to the supply $V_+$. When the other control voltage, $V_R$, is applied to the right actuator, the beam snaps back and switches the output back to 0 V, resetting the OMS ready for the next cycle of operation. The corresponding voltage change of a typical cycle of the operation of the OMS is presented in Fig. 2b. The output, $Q$, shows a lag behind the control voltage, $V_S$, which is denoted as gate delay. With 2.4 V of control voltage, the gate delay is -1.5s. The gate delay mainly depends on the complicated interaction of mechanical, thermal, and electrical properties of the OMS; which can be captured by a simplified analytical model (see Supplementary Note 7). For an OMS device with fixed geometry and materials, the delay is primarily determined by the amplitude of the voltage $V_S$ and cooling conditions. We studied the dependence on these two factors (Fig. 2c). We first placed the OMS in the still air and varied the amplitude of the control voltage, $V_S$. By monitoring the output voltage of the OMS, we can characterize the gate delay as a function of the control voltage $V_S$ (Fig. 2c).

When the voltage was small, the actuator could not heat up to sufficient temperature to drive the bistable beam snap-through. As the voltage increased, once it exceeded the threshold value, the OMS became functional; the delay dropped monotonically as the voltage increased further, approaching its lower bound, -0.1s (see Supplementary Note 7). By fitting the experimental results of the gate delay versus control voltage, we obtained the thermal mass, $C_{th}$, and absolute thermal conductivity, $\lambda$, of the actuator to be $4.48 \times 10^{-2}$ Ws/°C and $1.13 \times 10^{-2}$ W/°C, respectively. The fitting curve (the red dash line in Fig. 2c, $R^2 = 0.997$) indicates the lower bound of the voltage is around 1.19 V (Supplementary Eq. (10)), which is very close to our experimental

observation, i.e., 1.2 V. In addition, the fitting curve suggests that the period asymptotically approaches a certain lower bound (0.1 s) with increasing supply power; however, real limits on the delay include snap-through duration, inertial dynamics (e.g., air damping), and electrical contact formation. To further reduce the gate delay, we can implement a bistable beam with a smaller timescale (see Supplementary Note 8).

By adding a constant cooling air on the OMS, the voltage-delay curve remained similar with a voltage shift; the lower bound increased to 1.8 V (black data point, Fig. 2c). The fit curve with cooling air (black dashed curve, $R^2$ = 0.997) matches well with the experimental data, indicating the thermal mass, $C_{th}$, and absolute thermal conductivity, $\lambda$, of the actuator to be $4.62 \times 10^{-2}$ Ws/°C and $2.58 \times 10^{-2}$ W/°C, respectively. The model suggests a lower bound of around 1.80 V control voltage, aligning well with our experimental observation, i.e., 1.8 V.

Since the input circuit and control circuit are independent, the OMS can be used as a relay. Figure 2d shows the response of the switch to 10-s-long voltage pulses of $V_+$ = 2.4 V as the input signal and supply voltage rise to 9.6 V. The OMS relay can control the output with a voltage up to 4 times that of the control signal. This relay can also be used to control outputs with much higher voltages due to the electrical isolation of the input circuit from the control circuit.

The hysteresis of the beam makes the operation of this binary switch robust to noise. Noise in the control signal will not transmit to the output when it is moderate compared to the critical control voltage (i.e., 1.8 V). To demonstrate this property, we applied three types of voltage pulses of $V_+$ = 2.4 V to the left actuator (Fig. 2e). The first signal is a 20-s-long constant voltage of 0 V accompanied by moderate noise. The output signal did not change since the control signal with noise was below the critical voltage. The second signal is a pulse with a maximum voltage amplitude of 2.4 V with noise added during the off-state. The noise did not affect the output until the control signal reached the critical switching value. The last signal is a pulse with a maximum voltage amplitude of 2.4 V with noise superposed during the on-state. After the output switched to its high voltage state, the noise did not alter its binary voltage value afterwards. It is worth noting the actual values of all outputs in Fig. 2e changed slightly when noise was introduced. This is mainly caused by the compliance of OMS devices; fluctuation in control signals could induce deformation on both the bistable beam and the base of the OMS, which leads to the variation of the pressure on contact pads and thus results in undulation in output signals. However, these tiny ripples would not change the binary output voltage values.

For continuous operation, the gates need to be reset after each computational execution. This reset not only includes toggling the bistable beam back to its initial stable equilibrium but also requires bringing down the actuators to ambient temperature. Thus, the delay should be more clearly defined as the time taken for a fully reset gate to change its output upon input. Therefore, the switching speed (time duration between two executions) of such a gate needs to be sufficiently large to compensate for the cooling time required for actuators. As shown in Fig. 2c, we can achieve a minimal gate delay of ~0.1 s, which indicates a maximum switching speed of ~3.2 s with a cooling time of ~3 s.

### Origami digital computation and memory bit
**Fundamental logic gates.** Based on the OMS, we can realize a functional-complete digital logic system that includes all three fundamental logic gates: NOT, AND, and OR. Here, we assigned voltage 3.0 V (for driving CSCP actuators of the display) the binary logic value '1' and voltage 0 V the binary logic value '0'. Note that these gates need to be reset every time for subsequent operation.

An Origami NOT gate was designed by configuring $V_-$ as '1' and $V_+$ as '0', which provides the negation of the input signal $A$, i.e., $V_S$ (Fig. 3a). $V_R$ is assigned as reset $R$. Only when input $A$ is '1', the output of NOT

changes from '0' to '1' due to the snap-through of the bistable beam. We used a customized display to visually indicate the output of the gate (Fig. 3b). The display shows '1' when the output of the gate is '1', and vice versa (see Supplementary Movie 1).

Similarly, both AND and OR logic gates can be built by configuring the input voltage connections based on the OMS (Fig. 3c). The switch is configured as an AND gate by assigning $V_S$ as $A$, $V_+$ as $B$, and $V_-$ as '0'. In this configuration, only when both inputs $A$ and $B$ are '1', the gate outputs '1'. The OR is constructed by rearranging the inputs and connections of the OMS in a similar manner; the OR gate will only output '0' when both $A$ and $B$ are '0'.

**Combinational logic gates.** So far, we have successfully implemented single logic gates, i.e., NOT, AND, and OR gates, which provide the basis for a functionally complete set of logical connectives. However, to create more complex functions, it is necessary to compose multiple logic gates. Thanks to the cascadable configuration and low internal resistance, we could compose several OMSs directly without requiring any intermediates, which greatly reduces the complexity, fabrication difficulty, and energy consumption of resulting systems. For example, we can construct composite logic gates, i.e., NAND and NOR, by integrating two OMSs (Fig. 3d). We demonstrated the logic operation of all the mentioned logic gates, including AND, OR, NAND, and NOR, experimentally with all four possible inputs (Fig. 3e, see Supplementary Movie 2–5 for the full demonstration of AND, OR, NAND, and NOR, respectively). The remaining two basic logic gates, i.e., XOR and XNOR gates, can also be built easily in the same manner (see Supplementary Figs. 11 and 12), although we have not implemented them experimentally in this paper. More complicated combinational circuits, e.g., a half adder, are possible by combining multiple logic gates. The ability to compose multiple logical functions into a more complex circuit enables the exploitation and integration of various sophisticated computation and control in digital electronics and robotics.

**Nonvolatile memory bit.** Nonvolatile memory usually contains crucial programs of operation and can sustain power outages, making it essential components for autonomous control of robots. We built a simple origami Set-Reset latch with permanent storage capability upon the bistable switch design. The schematic of the S-R latch is shown in Fig. 4a, where $V_+$ = 1, $V_-$ = 0. Meanwhile, $V_S$ and $V_R$ are reconfigured as SET and RESET, allowing the writing and erasing of information inside the memory device. A detailed demonstration is shown in Fig. 4b: After supplying power to the device, we wrote a bit '1' after 6 s. After about a 1.5 s delay, the output $Q$ was modified to reflect the '1' input; this delay is the hold time of the latch. Then we intentionally introduced a power outage, causing the output to drop to '0'. However, we could still read the stored information after power was recovered. We could also delete/reset the memory (back to '0') by activating $R$ with supplied power; the low voltage memory is not affected by the power outage as well. One memory device is capable of storing one bit, i.e., two states. With more memory units integrated, it allows for the storage of $N = 2^n$ states where $n$ is the number of bits.

### Integrated autonomous robots with OMSs
To demonstrate the potential of the OMS in intelligent compliant devices and robotics, we used OMS-based components to control three origami robots: a flytrap-inspired robot that can autonomously sense, decide, and respond to environmental stimuli, i.e., physical touch (Fig. 5), an untethered self-reversing legged robot that can detect obstacles and reverse its locomotion direction (Fig. 6), and an origami wheeled car that can move with prescribed trajectories by utilizing reprogrammable origami memory (Fig. 7).

**Flytrap-inspired prey-catching robot.** The *Venus flytrap* is a carnivorous plant that is capable of distinguishing between living prey and

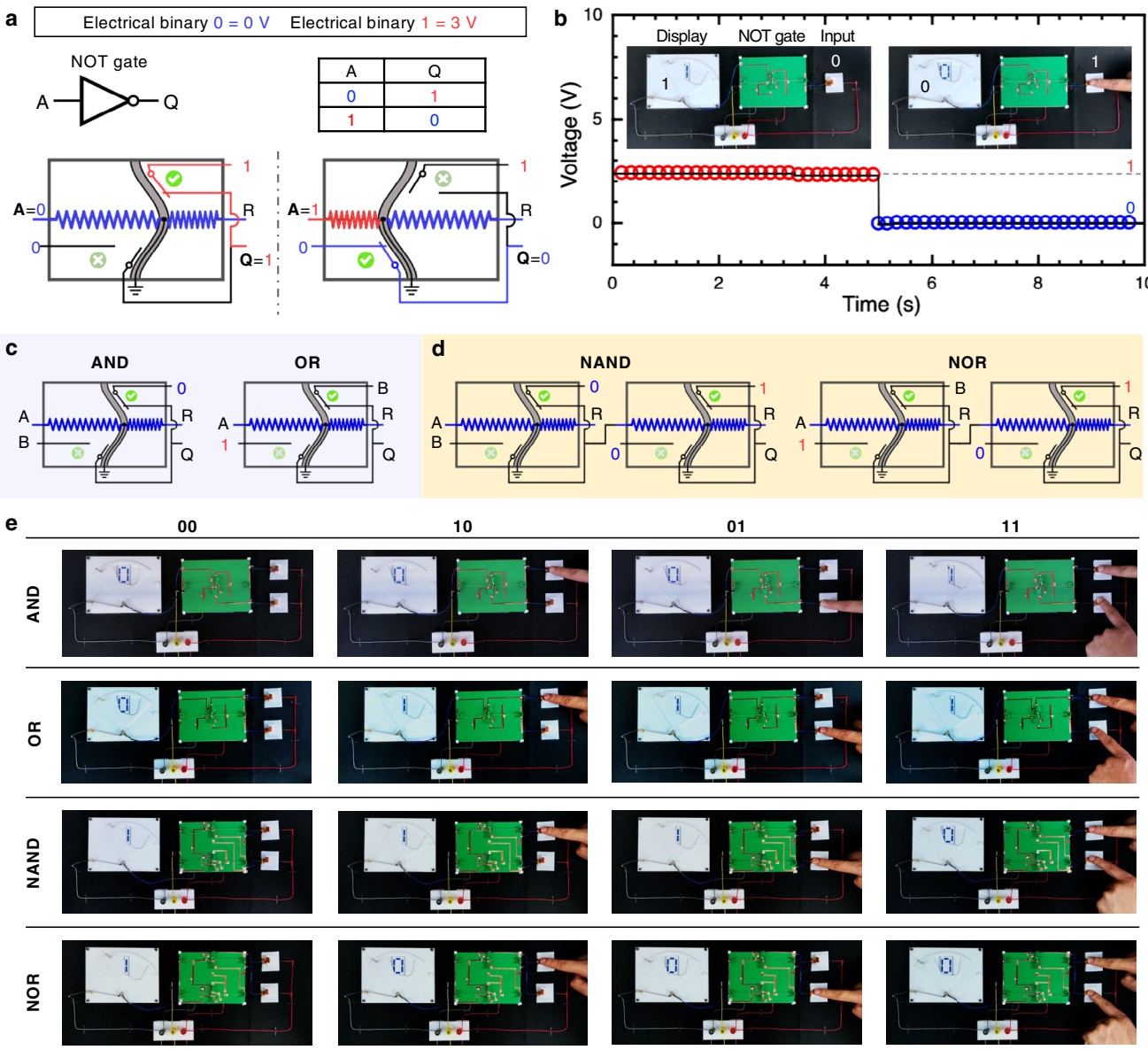

**Fig. 3 | OMS-based digital logic gates. a** An origami NOT gate. Logic diagram and truth table of a NOT gate. The schematic of an origami NOT gate in our architecture. **b** The input of the NOT gate is supplied by an origami switch; the output of the NOT gate is used to drive a CSCP actuator to change the reading ('0' or '1') of a display. **c** By configuring OMS, the other two fundamental logic elements, i.e., AND and OR gates, can be created. **d** Cascaded logic gates, i.e., NAND and NOR gates, could also be built by compositing two OMSs in series with specific configurations. **e** The full truth tables are demonstrated for the remaining logic elements, including AND, OR, NAND, and NOR gates.

non-prey stimuli (Fig. 5a). The leaves (or trap) only close when there have been two stimuli of the trigger hairs within approximately 30 seconds; this closing strategy is to avoid inadvertent triggering of the trap by inanimate objects, e.g., fallen leaves, to save substantial energy. Here, we constructed a flytrap-inspired robot (Fig. 5b, see Supplementary Note 2 for more details) that partially imitates the prey strategy based on the proposed OMS. Although our robot does not consider the temporal information contained in stimuli, it can still be used to capture small 'living prey' since an immobile object cannot activate both sensors. However, without including temporal information from stimuli, our robot cannot specify desired preys with a certain moving speed as its biological analogs[49]. Meanwhile, the robot can selectively catch a large prey since a small one is difficult to detect with both sensors. This robot consists of two origami touch sensors, one origami AND gate, and two CSCP actuators (in parallel). The schematic of the robot is detailed in Fig. 5c. It uses touch sensors to receive

stimuli from the environment (Fig. 5d), which are then passed to the controller, i.e., an AND gate, for analysis. This results in an executable signal downstream to the CSCP actuators to control the open/closed states of the leaves. Only when two sensors are activated, the flytrap-inspired robot can 'recognize' it as a living prey and 'decide' to close its leaves to capture it by contracting CSCP actuators; otherwise, the CSCP actuators are kept inactivated with the leaves open (Fig. 5c). The touch sensor is modified from a bistable beam as shown in Fig. 5d, where a touch can trigger the snap-through of the bistable beam to change the on/off status of the circuit on it. Each sensor is integrated into the inner surface of two leaves (Fig. 5b). Two leaves are connected on the top surface of the AND gate through origami tabs, which function as hinges for the motion of the leaves when driven by actuators. One end of each CSCP actuator is fixed on the bottom of a leaf while the other is attached to the support structure. Actuators are deployed diagonally for maximal actuation stroke (Fig. 5e).

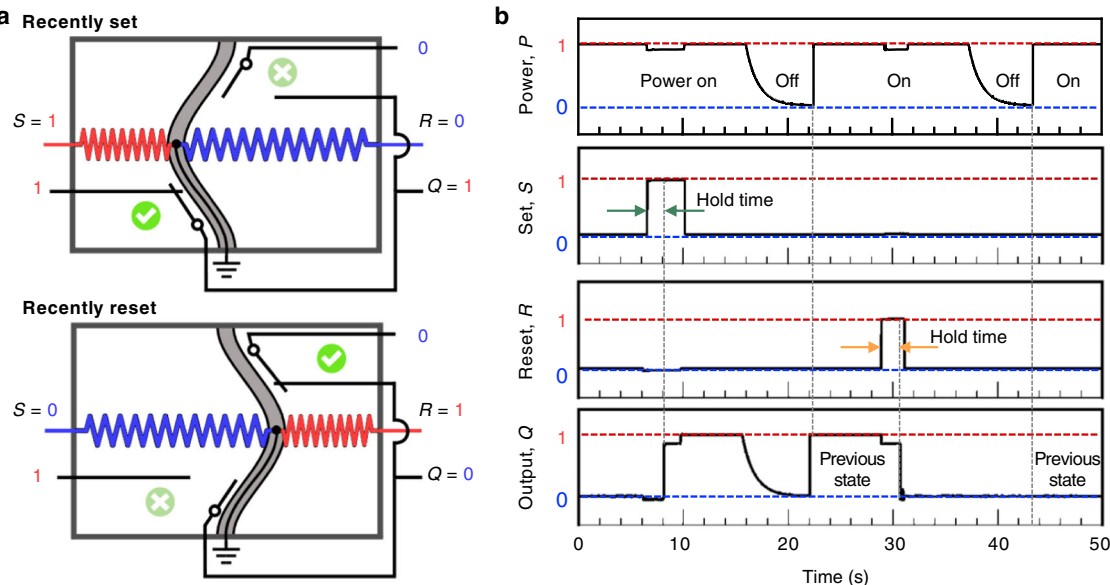

**Fig. 4 | Origami memory bit. a** An OMS is reconfigured as a set-reset latch for information storage. The S and R represent SET and RESET. If the latch is set, it outputs high voltage '1'; if reset, it yields low voltage, '0'. **b** Demonstration of writing, erasing, and nonvolatile memory of the latch with a featured hold time. The latch can sustain its recently written information after a power outage.

To demonstrate, we used the flytrap-inspired robot to capture a living 'prey' according to the interactions with an object (see Fig. 5f–i). Initially, the leaves of the robot were open. Though the object fell into the trap, the robot did not close its leaves if the object did not touch any sensor (Fig. 5f). Correspondingly, the output voltage of the circuit is 0 V, which indicates no contraction of the CSCP actuator. Even if the object activated one of the sensors, the robot would identify it as non-prey stimuli since single touch indicates immobility (Fig. 5g, h). Once the object is capable of triggering both sensors in two different positions, the robot would assume the object is moving 'prey' and close its leaves to capture it (Fig. 5i and see Supplemental Movie 7 for the full demonstration). The voltage signals of sensor A, sensor B, and the CSCP actuator of the flytrap-inspired robot also suggests a successive capture with a delay of ~1.7 s (Fig. 5f–i). The robot usually takes several seconds (e.g., 5 s in this case) to close its leaves due to system delay and low supply voltage (i.e., 2.4 V). Though the closing speed is out of the scope of this work, we can potentially improve it by two methods. Firstly, we can increase the length of the CSCP actuators with higher supply voltage which would increase the actuator speed[45]. Otherwise we can incorporate mechanical snap-through mechanisms that would further reduce the closing time, possibly down to 100 ms[50]. In this work, we chose our design based on clarity and simplicity.

The flytrap-inspired robot is fully fabricated through origami-inspired cut-and-fold out of sheet materials and conductive threads (except the 3D printed support structure i.e., its environment, though it too could be implemented in origami), which leads to semiconductor-free and nonmagnetic features, suggesting broad applications, especially in extreme environments, such as high radiation/magnetic fields, where typical semiconductor-based electronic components could not function[1]. Specifically, we operated the origami-based robot under four adversarial environmental events, i.e., static magnetic field (0.47 T), radio frequency (RF) signal (power: 5 W), electrostatic discharge (ESD, output voltage ≥20 KV), and mechanical deformation (up to 50° bending and twisting). The results show that our robot could perform the designed task robustly (Fig. 5j) while its semiconductor-based counterpart malfunctioned or even failed permanently (Supplementary Movie 9–12). Although not directly tested, radiation damage to semiconductor-based devices has long been identified and well-researched[51], which also indicates

the advantage of our approach in such environments. Moreover, our origami-based robots compare favorably against their semiconductor-based analogs, especially in terms of weight and cost, quantifying the benefits of our method (Supplementary Table 3). In addition, there are applications where the delicate touch and intelligent capture of a soft gripper are desired, e.g., sampling brittle sea animals[52]. In most cases, these grippers require external manual control from operators to execute a capture order. Instead, our robot could be used as a smart alternative to automatically recognize and capture living fragile animals without requiring human intervention or additional decision-making components; this new strategy can potentially simplify system complexity and improve operation robustness to semiconductor-based electronics unfriendly environments.

**Untethered self-reversing legged robot.** Collision avoidance is one of the most essential needs of biological agents when exploring the environment, which is achieved by collecting information and thus responding upon analysis. For example, cockroaches rely on tactile sensing for perceiving physical objects to explore a neighborhood since most cockroach species are nocturnal[53]. Specifically, a cockroach can achieve collision avoidance by sensing obstacles with the antennae on their heads. To demonstrate our method, we designed an untethered self-reversing legged robot that can reverse its locomotion upon detecting obstacles inspired by the behavior of collision avoidance of cockroach (Fig. 6a).

This cockroach-inspired untethered legged robot is mainly composed of two touch sensors A and B (with the corresponding antenna), one on-board origami controller, two modified DC motors, and one lithium battery (Supplementary Note 3). The simplified schematic of the robot is presented in Fig. 6b. The touch sensor consists of two initially disconnected copper strips, which could be forced closed upon the collision of the antenna to transit voltage signal to the on-board origami controller. This origami controller consists of an OR gate, a CSCP actuator, and a double-pole double-throw (DPDT) switch (Fig. 6c). This DPDT switch is modified from the OMS by adding another set of circuits on the bistable beam. This switch controls the current flow direction between four ports by toggling between two different states

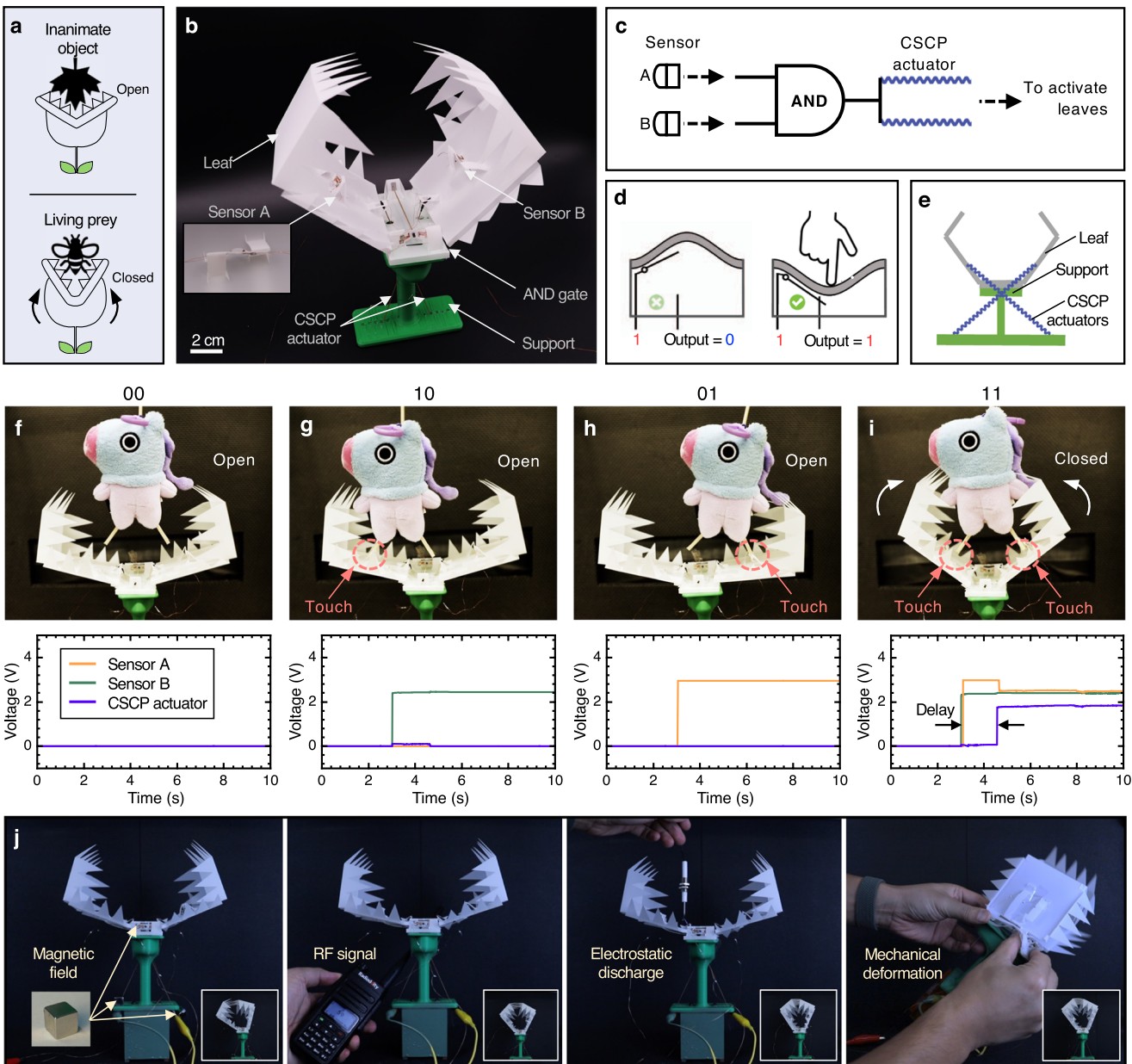

**Fig. 5 | The origami flytrap-inspired prey-catching robot. a** *Venus flytrap* can distinguish living preys from inanimate stimuli and close its leaves to capture preys. **b** The detailed structure of the robot with labels. **c** The simplified circuit diagram of the robot. Two touch sensors located on the leaves perceive external information and send that information as a voltage signal to the AND gate; the output (voltage) of the gate, in turn, is used to indicate the operation of the CSCP actuators that are used to change the state of the leaves. **d** The schematic of a touch sensor modified from a bistable switch. **e** The actuation system of the flytrap-inspired robot. Two CSCP actuators are attached to the two leaves, respectively, to supply actuation to close them. **f–i** Demonstration of a living `prey' capture. This robot would only close its leaves when it sensed two touches from the two targeted locations on the leaves, indicating the falling object is a living 'prey'. Top: snapshots of key image frames; bottom: voltage of the outputs of two sensors and across the CSCP actuators that are used to activate the motion of the leaves. **j** The origami flytrap-inspired robot could survive a strong magnetic field (0.47T), intense radio frequency signal interference (power, 5W), high electrostatic discharge (output voltage ≥20 kv), and large mechanical deformation (up to 50° bending and twisting) while its semiconductor-based counterpart malfunctioned under the same conditions. More details can be found in Supplementary Movies 9–12.

(corresponding to counterclockwise or clockwise rotation of the legs.) Once the OR gate is triggered by the signals from sensors (and antennae), its high output voltage will drive the CSCP actuator to toggle the DPDT switch to change the rotating direction of DC motors to reverse the locomotion (Fig. 6b). Figure 6d–g shows key frames of the behavior of the legged robot in different environments. When there is no obstacle, the robot can continuously move forward. However, the robot would reverse its direction when either antenna (representing '10' or '01' for inputs of the OR gate) or both

antennae (representing '11' for inputs of the OR gate) detect obstacles (see Supplemental Movie 6 for the full demonstration).

The sensing, computation, and control were done on-board the robot, demonstrating a system with semi/autonomous behaviors that can be integrated into the body of an origami robot. There are various applications where semiconductor-free self-reversing robots are of special interest. For example, the resulting crawler is of great potential for tasks, such as exploration and rescue, in extreme areas (e.g., high radiation fields).

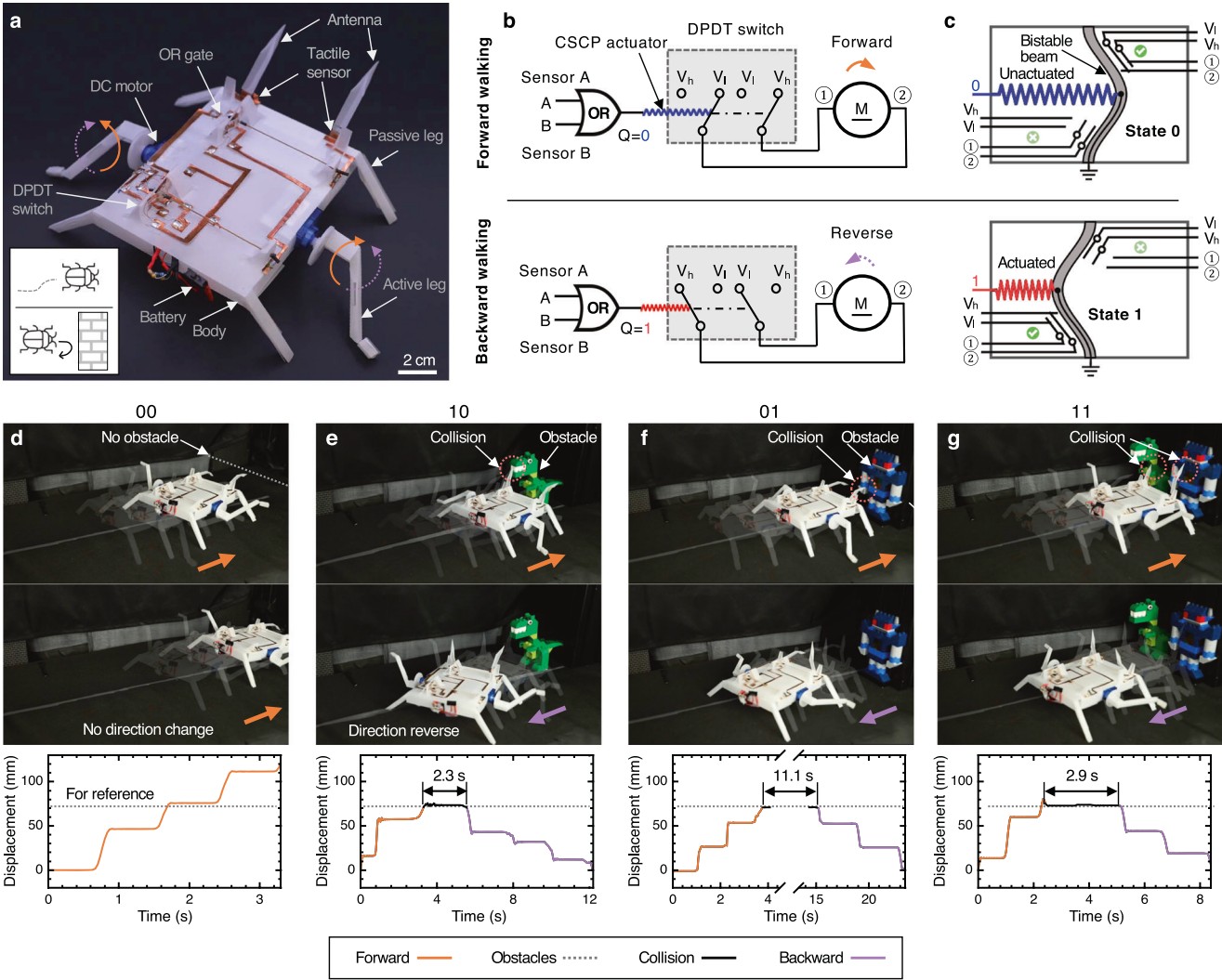

**Fig. 6 | The cockroach-inspired self-reversing legged robot. a** The detailed structure of the legged robot with labels. **b** The simplified circuit diagram of the legged robot. The information from the two tactile sensors on antenna decides the output of the origami OR gate; the output of the gate is used to determine the actuation of the CSCP actuator, which could potentially change the rotation direction of DC motors through a DPDT switch. **c** The schematic of an origami

DPDT switch modified from the OMS. The switch of the states of the bistable beam changes the direction of rotation of the motors between counterclockwise and clockwise. **d–g** The legged robot encountering obstacles. Top: overlaid sequential images shows the crawling direction; bottom: displacement curves of the robot. Images and displacement curves are both derived from videos (see Supplementary Movie 6).

Although the periodic actuation of the legs could have been generated by origami oscillators themselves[25], in this instance we used conventional components (in the form of DC motors used to generate motion), demonstrating the general interfacing allowing our systems to co-exist in existing robot ecosystems.

**Untethered origami car with reprogrammable trajectories.** Robots that autonomously locomote along specified open-loop trajectories can serve as platforms for practical applications, e.g., detecting hazardous leakages with gas sensors equipped or executing a surveillance function with cameras on a well-defined route[54]. Here we design an origami wheeled car (Fig. 7a and Supplementary Note 4) that can follow designed trajectory patterns by reading from an origami memory disc composed from OMSs; a sequence of information (bits) written in the disc is extracted by the rotating read head and then sent to two DC motors to propel the origami car. By varying the memory, the car can drive along different prescribed trajectories.

The disc is mainly composed of four composite memory words, i.e., |**a**|**b**|**c**|**d**|; each word consists of two memory bits to control the corresponding pair of motors, respectively (Fig. 7b). For example, the

word **a** contains $a_l$ and $a_r$ (i.e., **a** = $a_l a_r$); $a_l$ is to control the left motor ($M_l$) for driving the left wheel while $a_r$ is for the right one through $M_r$. When the written information in the memory bit is '0' (representing low voltage, 1.5 V), the corresponding wheel rotates slow; otherwise the wheel spins fast from memory '1' (high voltage, 3.0 V). The basic memory bit (Fig. 7c) is modified from the origami Set-Reset latch (Fig. 4). The origami car has four fundamental locomotion modes due to the combination of two basic memory bits (Fig. 7d). When both wheels receive the same information, the car moves straight forward (if both '0', the car moves slowly; if both '1', it locomotes fast). Otherwise, the car would either turn left ('01') or right ('10') to change the locomotion direction.

To demonstrate, we programmed the origami wheeled robot to follow the locomotion paths of the letters 'u', 'c', 'l', and 'a', respectively, to illustrate its trajectory specification (Fig. 7e). The X-Y position information of the trajectories was extracted from the video (Supplementary Movie 8) and shown in Fig. 7f with the time mapped as color. Between operations, the memory in the disc needs to be erased and rewritten by switching the states of the bistable beam of the basic memory bits. For example, the memory was set to be |**a**|**b**|**c**|**d**| =

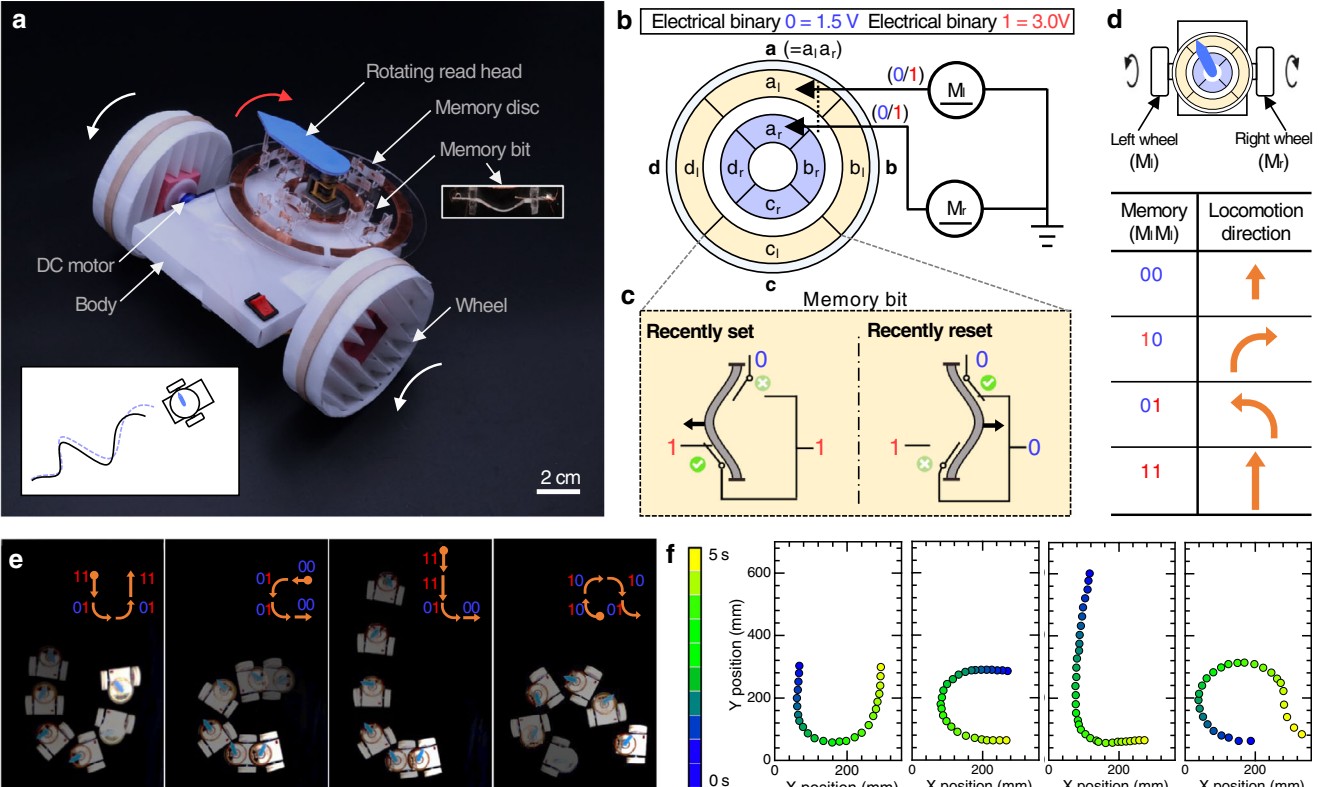

**Fig. 7 | Origami wheeled car. a** The detailed structure of the origami car with labels. The rotating read head extracts the stored memory (voltage) from the disc and forwards that to two DC motors to control the locomotion direction of the car. **b** The simplified circuit diagram of the wheeled car. Each composite memory bit consists of two basic memory bits, which control the corresponding wheels through two DC motors. For example, composite memory bit, a, is composed of two basic memory

bit, $a_l$ and $a_r$; $a_l$ is used to control the left wheel ($M_l$) while $a_r$ is for the right one ($M_r$). **c** The schematic of a basic memory bit. **d** The locomotion direction of the car is determined by the composite memory bit; the trajectory of the car is controlled by the sequence of memory bits. **e** Overlaid sequential images (derived from video frames) visualize the trajectories of the car (programmed with different locomotion plans: 'u', 'c', 'l', and 'a'. **f** X-Y position of the car in **e** (see Supplemental Movie 8).

|11|01|01|11| for the path of 'u'; it was reprogrammed as |00|01|01|00| for that of 'c'. In the same manner, the memory was modified as |11|11|01|00| and |10|10|10|01| for 'l' and 'a', respectively.

Currently, the switching between two states of the basic memory bit is manually manipulated for simplicity. The automation of the memory writing could be done by incorporating programmable controlling mechanisms, e.g., CSCP actuators (actuated by voltage signals, Fig. 4a) or through magnetic fields (with magnetic beads on the bistable beam)[55]. The origami car can have 256 (=$2^8$) different trajectories based on the current design of the memory disc (four 2-bit memory words, i.e., 8 memory bits in total). To achieve locomotion with more sophisticated trajectories, we could increase the physical density of the memory on the disc to expand the number of words in the memory bank, increasing the complexity of the representable trajectories. Otherwise we could increase the depth of the memory, extending each word to include more bits, e.g., $a_d \rightarrow \mathbf{a} = a_l a_r a_d$ to control the rotation direction (clockwise or counterclockwise) of the motors and thus the overall locomotion direction (forward or backward) of the car. Such additional bits could be integrated through the origami DPDT switches to control the flow of current through the actuators (Fig. 5c). When further equipped with sensing capabilities, the origami car could presumably execute meaningful tasks such as route surveillance and radioactive leakage detection in nuclear power plants. The dependence on a DC motor to drive the memory read head could be eliminated by using linear sliding instead of rotating brushing; for example, a head (driven by a linear actuator, e.g., CSCP actuator) could slide over a memory tape to extract the stored information.

## Discussion

This work proposes an integrated process for creating autonomous robots with sensing, control, and actuation directly embedded into compliant origami materials and structures. This method is enabled by our origami multiplexed switch (OMS) that functions as a multiplexer by harnessing a snap-through instability to control electrical signals. The unique design endows OMS with multiple functions. The OMS can be used as a relay and functions robustly to noise, which are validated through experimental demonstrations. The OMS can also be reconfigured into origami logic gates (including NOT, AND, and OR, with functional completeness). We have further shown that it is practical to compose multiple OMSs in a cascading manner by using NAND and NOR gates as examples. This successful composition suggests that the design space of our origami computing architectures is far larger than those presented in this paper. In addition, to demonstrate the potential of the OMS for origami robots, we built a 1-bit storage device (i.e., Set-Reset latch) from an OMS that can write, erase, rewrite itself, and sustain power outage, which would be necessary for achieving a higher-level autonomy of robots. To achieve interaction with environments, we further proposed origami sensors and constructed a complete sense-decide-act loop, which is illustrated with three autonomous origami robots. Through these, we demonstrate a solid step towards the construction of untethered and autonomous origami robots, as well as intelligent matter, using only cut-and-fold processing. More specific discussions can be found in Supplementary Discussion.

In summary, our proposed method provides an alternative way to create autonomous origami robots by tightly integrating intelligent conductive materials to achieve a wide range of functionalities. These functions include self-sustained locomotion and sequencing through

oscillation[25], information processing, logic and computing, and various human/environment-machine interactions. Our method has demonstrated the great potential to enable origami functional materials and mechanisms to further advance the autonomy of robotics (e.g., realizing a finite state machine or even a Turing machine) while preserving the advantageous features inherent in origami-based fabrication (see Supplementary Table 4). Our work also provides a versatile platform that could motivate interdisciplinary innovation in design, mechanics, materials, and fabrication.

## Methods

### Fabrication of CSCP actuators
The CSCP actuators were formed by using commercially available conductive yarn (235-34 4ply HCB, V Technical Textiles Inc.) with a diameter of ~0.4 mm. These actuators were prepared by following three steps (Supplementary Fig. 2). We firstly insert coils by continuously twisting the conductive yarn under a 280-gram weight by a stepper motor (XY42STH34-0354A, Guangzhou Shenglong Motor Co. Ltd.). The weight was free to move vertically but not allowed to rotate. We then anneal the coiled yarn with a cycling heating/cooling process (0.45 A annealing current, 30 s heating, and 30 s cooling per cycle, 8 h). The last step is to stabilize the actuator by repeating the heating/cooling process (0.27 A training current, 10 s heating, and 10 s cooling per cycle) without tension for 5 min. The obtained actuators have an average diameter of ~0.71 mm.

### Fabrication of origami devices
Origami devices are mainly composed of folded bodies, contact pads, copper circuit traces, and CSCP actuators. The origami bodies or farmings were folded manually from a patterned 0.127-mm-thick flexible, polyester film (DuraLar™, Grafix Plastics). The pattern was cut with a cutting machine (Silhouette CAMEO 2, Silhouette America, Inc.). The connections of device between substructures were jointed by origami tabs to minimize required resources. The contact pads and circuit traces were made out of copper tape (0.05 mm, Elite O&S), which was patterned and cut by a laser machine (Speedy 300™ Flexx, Trotec Laser Inc.) or by a craft knife manually on customized masks. CSCP actuators were then assembled to complete the design. Specifications to the fabrication of all devices, along with corresponding fabrication 2D patterns, are detailed in the Supplementary Note 1–4.

### Characterization of the OMS and robots
The devices were powered through a laboratory DC power supply (TP-3003D-3, Kaito Electronics, Inc.). The current and voltage are recorded by using NI myDAQ (National Instruments). The displacement of devices was characterized from a digital camera (30 fps) respectively. To decrease the cooling time on the actuator, an active cooling system was implemented with the default parameters set to ~22 °C (room temperature), ~110 cfm flow rate, ~130 mm distance from the tested devices. Other variations of these parameters were implemented for various applications.

## Data availability
All the data supporting the finding of this study are available within the main text, its Supplementary Information, and Supplementary Files or from the corresponding authors upon request.

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

## Acknowledgements

We acknowledge National Science Foundation (NSF) award no.1752575 received by A.M. for supporting this research. We thank Mr. D. Martinez and Mr. W. Fernando for assisting with the fabrication and testing of flytrap-inspired robots. We also thank Dr. R. Wood for the feedback on manuscript preparation.

## Author contributions

W.Y. conceived the concept; W.Y. designed the research; W.Y. conducted the device fabrication and characterization; W.Y., S.L., and M.D. designed and conducted robotic demonstrations; W.Y. and Z.Z. analyzed the data; W.Y. prepared figures and drafted the manuscript; W.Y., S.L., D.R., and A.M. revised the manuscript; D.R. and A.M. supervised the research.

## Competing interests

W.Y. and A.M. are inventors on a provisional patent application on the presented technology. The other authors declare that they have no competing interests.
