## [Peer Review File · Nature Communications]

Origami-based integration of robots that sense, decide, and respondREVIEWER COMMENTS

Reviewer #1 (Remarks to the Author):

This manuscript introduces an origami multiplexed switch based on bistable beams and thermal polymeric actuators. The switch is characterised and then used to build fundamental computational structures including logic gates and an SR latch. Potential applications for this approach are demonstrated by using integrated OMS structures to control three autonomous robots.

The OMS structure is elegant and to the best of my knowledge origami structures for computation have not been previously demonstrated, meaning this work is likely to be of interest to the field of origami robotics. This work is also complementary to a number of recent works which explore alternative methods of computing, as referenced in the manuscript.

I think the authors could present a stronger case for why non-semiconductor based control is beneficial for origami robots; a number of reasons are listed, but none of them are particularly convincing.

The authors' main conclusion is that their device represents a route towards autonomous origami robots by demonstrating a functional sense-plan-act control loop and its integration into a number of autonomous robots. I feel like this conclusion is supported by the demonstration of OMS-based devices and their integration into simple autonomous robots. I have some concerns about how much further this approach can be taken, but the authors have discussed potential limitations due to fabrication and scale of their device.

More specific concerns are listed below, but overall, I feel like this work is novel, clearly presented and worthy of publication.

=====

1. The authors claim that the low resistance of the OMS enables the possibility of creating more complex circuits. I agree that the resistance is low enough for this, but I wonder about the gate delay -- given a switching time of 1.5 seconds, how many OMS based devices can be chained together before the gate delays becomes impractical? I think this point could be explored a little more.

2. For the characterisation of gate delay, I'm wondering if the delay remains constant throughout operation? I'd assume that the delay would be higher at faster switching rates, as the CSCP actuator would have to also work against the other CSCP. Looking at figure 2B, it looks like the gate delay for the second switch (V_r) is slightly longer than the first switch (V_s). I think the authors need to include further discussion around this point -- it would be interesting to report a maximum switching speed as well as the gate delay.

3. I'm not sure if referring to the OMS as an amplifier is correct. I'd expect an amplifier to have some kind of relationship between the input and output (i.e a bit like operating a transistor in its linear region).

4. I found the explanation of the noise filtering part a little unclear. The manuscript says that the noise was filtered, but it appears that the noise *did* cause the output to change in the

middle and right panels? I think the figure needs to make it clear what the ideal control input should be, so we can judge what the expected output should be. The text refers to three types of voltage pulse, but I'm not sure what these are.

5. For the logic gate section, I think the author's need to be clearer that their gates need to be reset between cycles. This isn't something I would expect, and although the author's briefly mention that V_r is used as a reset, I think it needs to be discussed a bit more. I appreciate that the author's include this in the discussion section and that it could be addressed with a monostable beam, but I think it should be made clear in this section too as it's an important limitation.

6. I'm not convinced by the claim that the first demonstration truly mimics the Venus flytrap's ability to capture 'living prey'. I'm far from an expert in carnivorous plants, but I think the venus flytrap requires its trigger hairs to be activated with a short delay between them? I don't think this temporal aspect is included in the demonstration -- it might be better to say that the demonstration selects for large prey (as smaller prey would only activate one of the sensors).

Reviewer #2 (Remarks to the Author):

This manuscript reports an integrated fabrication process for fabricating origami robots with memory bits, touch sensors, digital logic gates, and cascaded circuits. The origami robots were capable of achieving a complete sense-decide-act loop to realize autonomous interaction with environments. Several origami robots were fabricated and demonstrated in this manuscript. Although this paper has reported a new approach to fabricating autonomous origami robots, I still have several concerns and comments on this work, which are listed below. I will suggest this paper for publishing in a more robot-specific journal, considering the general audience in Nature Communications.

1. First, as the authors mentioned in the Introduction, there are several challenges in using rigid semiconductor-based electronics. For instance, semiconductor-based electronics are typically vulnerable to environmental factors, e.g., radiation and physical impact, which limit their applications. The authors should demonstrate that their origami robots can endure various environmental factors (radiation and physical impacts), proving their approaches with clear advantages over the origami robots with semiconductor-based electronics.

2. In the Introduction, the authors also mention that equipping external semiconductor-based electronics requires system integration that would increase the complexity and weight. The authors should make head-to-head comparisons of robotic weights and energy consumption between their origami robot systems and conventional origami robots with semiconductor-based electronics.

3. From Figure 1c–1e, most of the control units and electric power devices were installed on planar and non-actuatable locations. What are the advantages of the proposed cut-and-fold processing in comparison with the post-installation approach? I believe that conventional origami robots also install semiconductor-based electronics on planar and non-actuatable locations. Then, it became unclear to me the major advantages of the reported work.

4. Besides the artificial muscle actuators, can the authors' origami robot adopt other actuation systems, such as pneumatic systems or shape memory alloys?

5. The authors should provide a table to summarize the performance and properties of origami robots and soft robots in the literature, and the readers can clearly understand the

technological advances of this work.

Reviewer #3 (Remarks to the Author):

This paper describes a mechanical computing device that exploits buckling to achieve bistable behaviors and act as a switch. The so-called Origami Multiplexed Switch (OMS) can be manufactured by cutting and folding of paper, self-adhesive copper strips, and conductive thread.

The bistable beam can be actuated by two "fishing line" actuators and switches input current between two outputs, depending on its position. The authors show how these building blocks can be configured to implement basic logic gates, including NAND, which enables Turing universal computation.

While the OMS falls into a larger class of mechanical computing devices, this contribution is particularly interesting as the OMS integrates well with sensors and actuators, thereby allowing the realization of complete autonomous systems. This is demonstrated with three origami robots: a venus flytrap that requires two distinct physical contacts, a vehicle that reverses direction upon contact, and a second vehicle whose trajectory is programmed by a series of set-reset memory devices. Here, physical forces onto the robot translate into activation of a bistable beam and hence trigger the programmed behaviors. Although the demonstrated robots are very limited, the authors make a compelling case that other "smart polymers" can be used for sensing and activating the bistable beams and/or replacing the fishing line actuators altogether. As such, the OMS concept might find a variety of applications in creating stimuli-responsive materials that go beyond purely reactive behavior but can exploit memory and computation.

Supplementary materials describe the manufacturing process, a detailed analysis of the OMS dynamics (with switching times in the order of seconds), and an exploration of the possible design space that OMS can enable together with other means of sensing and actuation. Methods used are described in sufficient detail, albeit relying on previous publications of the authors for some aspects of the manufacturing process.

Response letter for NCOMMS-22-36157

REVIEWER COMMENTS

Reviewer #1 (Remarks to the Author):

This manuscript introduces an origami multiplexed switch based on bistable beams and thermal polymeric actuators. The switch is characterised and then used to build fundamental computational structures including logic gates and an SR latch. Potential applications for this approach are demonstrated by using integrated OMS structures to control three autonomous robots.

The OMS structure is elegant and to the best of my knowledge origami structures for computation have not been previously demonstrated, meaning this work is likely to be of interest to the field of origami robotics. This work is also complementary to a number of recent works which explore alternative methods of computing, as referenced in the manuscript.

Response: Thank you for confirming the novelty and importance of our work.

I think the authors could present a stronger case for why non-semiconductor based control is beneficial for origami robots; a number of reasons are listed, but none of them are particularly convincing.

Response: We appreciate the reviewer pointing this out. We have conducted more experiments by using the flytrap-inspired robot as an example to show the advantages of our origami-based control over the semiconductor-based model in terms of robustness to environmental factors, including magnetic field, radio frequency (RF) signal, electrostatic discharge (ESD), and mechanical deformation. We also compared the complexity, weight, and cost of these two robots to support our claims. We have summarized these results in the 3rd paragraph (line 315-324) of Section: “Flytrap-inspired prey-catching robot”, the newly added Supplementary Text S9 (line 994-1036), and Supplementary Table S3.

The authors' main conclusion is that their device represents a route towards autonomous origami robots by demonstrating a functional sense-plan-act control loop and its integration into a number of autonomous robots. I feel like this conclusion is supported by the demonstration of OMS-based devices and their integration into simple autonomous robots. I have some concerns about how much further this approach can be taken, but the authors have discussed potential limitations due to fabrication and scale of their device.

Response: Thanks for recognizing the contributions of our work.

More specific concerns are listed below, but overall, I feel like this work is novel, clearly presented and worthy of publication.

Response: Thanks again for supporting our work for publication in Nature Communications. We appreciate your constructive comments. We have addressed your concerns point-by-point as shown below.

=====

1. The authors claim that the low resistance of the OMS enables the possibility of creating more complex circuits. I agree that the resistance is low enough for this, but I wonder about the gate delay -- given a switching time of 1.5 seconds, how many OMS based devices can be chained together before the gate delays becomes impractical? I think this point could be explored a little more.

Response: Thanks for asking. We agree with the reviewer that this issue should be discussed more. The response speed of both the CSCP actuators and the bistable beam essentially determines the gates' switching time or so-called gate delay. In the current configuration, the delay of the cascaded circuit is almost linearly proportional to the number of gates in series. Thus, our system only targets speed-insensitive applications. In addition, our origami-based process provides alternative methods to create complicated circuits. For example, we can directly design a self-sustained oscillator [1] instead of combining several OMS-based gates. On the other hand, we can shorten the response time of the actuator by increasing the supply voltage as shown in Fig. 2C to a minimum of about 0.1 s. The snap-through speed of the bistable beam is intrinsically determined by its geometry and material properties, which can be increased by using a shorter, stiffer, or lightweight beam, or increasing its thickness. However, it is

very challenging to further reduce the switching time to be comparable with that of semiconductor-based devices, which is also not the goal of this work. We have included related discussions in the 6th paragraph (line 485-496) of Discussion and Supplementary Texts S7 (line 929-981) & S8 (line 982-993) of Supplementary Materials.

2. For the characterisation of gate delay, I'm wondering if the delay remains constant throughout operation? I'd assume that the delay would be higher at faster switching rates, as the CSCP actuator would have to also work against the other CSCP. Looking at figure 2B, it looks like the gate delay for the second switch (V_r) is slightly longer than the first switch (V_s). I think the authors need to include further discussion around this point -- it would be interesting to report a maximum switching speed as well as the gate delay.

Response: Thanks for this valuable suggestion. The reviewer is correct. The delay will change at different switching speeds. For example, the delay will be larger if the opposite CSCP actuator is not fully cooled down, which is corresponding to a fast switching speed. To be more precise, we define delay as the time required for the gate to change its status from a fully cooled-down initial condition. Thus, there is a maximum switching speed that is equal to the minimal cooling (or recovery) time of the actuator plus two times the gate delay. According to the results, as shown in Fig. 2, the gate delay is about 0.1 s. In this work, the cooling time of the actuator back to room temperature is about 3 s. Therefore, the maximum switching speed is 3.2 s. To further increase the switching speed, more effective cooling methods, e.g., water cooling, could be used. Otherwise, other non-thermally driven conductive actuators, like dielectric

elastomer actuators, can be adopted. We have included more discussions in the last (8th) paragraph (line 204-211) of the section “The origami, multiplexed switch” and Supplementary Text S8 (line 982-993).

3. I'm not sure if referring to the OMS as an amplifier is correct. I'd expect an amplifier to have some kind of relationship between the input and output (i.e a bit like operating a transistor in its linear region).

Response: We thank the reviewer for posting this question. We agree that the OMS in Fig. 2D is better referred to as a “relay” instead of an amplifier. We have also updated the terminology accordingly as shown in the 6th paragraph (line 183-188) of the section “The origami, multiplexed switch”.

4. I found the explanation of the noise filtering part a little unclear. The manuscript says that the noise was filtered, but it appears that the noise *did* cause the output to change in the middle and right panels? I think the figure needs to make it clear what the ideal control input should be, so we can judge what the expected output should be. The text refers to three types of voltage pulse, but I'm not sure what these are.

Response: We thank the reviewer for bringing this up. To make it clearer, we have modified Fig. 2E and included more explanations for the control signal in the 7th paragraph (line 189-203) of the section “The origami, multiplexed switch”.

5. For the logic gate section, I think the author's need to be clearer that their gates need to be reset between cycles. This isn't something I would expect, and although the author's briefly mention that V_r is used as a reset, I think it needs to be discussed a bit

more. I appreciate that the author's include this in the discussion section and that it could be addressed with a monostable beam, but I think it should be made clear in this section too as it's an important limitation.

Response: We appreciate the reviewer's suggestion, which definitely helps clarify the design of the OMS. We have added the clarification in the first paragraph (line 216-217) of the Section: "Origami digital computation and memory bit".

6. I'm not convinced by the claim that the first demonstration truly mimics the Venus flytrap's ability to capture 'living prey'. I'm far from an expert in carnivorous plants, but I think the venus flytrap requires its trigger hairs to be activated with a short delay between them? I don't think this temporal aspect is included in the demonstration -- it might be better to say that the demonstration selects for large prey (as smaller prey would only activate one of the sensors).

Response: Thanks for bringing this up. Our robot is inspired by the *Venus* flytrap but not fully mimicking its exact mechanism since we have yet to be capable of considering the temporal information of stimuli. Without incorporating this temporal information, our robot cannot specify desired prey with a certain moving speed as its biological analogs [2]. However, we argue that our robot can recognize and capture a small "living prey" because a small immobile object cannot activate both sensors which are located separately from each other. As the reviewer suggested, our robot can also be used to select large prey since small ones are hard to trigger both sensors. We have modified our statement according to the reviewer's suggestion mainly in the first paragraph (line

269-278) of the Section: Flytrap-inspired prey-catching robot. Other small modifications, spread over the whole manuscript, are not specified here.

Reviewer #2 (Remarks to the Author):

This manuscript reports an integrated fabrication process for fabricating origami robots with memory bits, touch sensors, digital logic gates, and cascaded circuits. The origami robots were capable of achieving a complete sense-decide-act loop to realize autonomous interaction with environments. Several origami robots were fabricated and demonstrated in this manuscript. Although this paper has reported a new approach to fabricating autonomous origami robots, I still have several concerns and comments on this work, which are listed below. I will suggest this paper for publishing in a more robot-specific journal, considering the general audience in Nature Communications.

Response: We thank the reviewer for recognizing our contribution that we presented a new approach for fabricating autonomous origami robots. Although rooted in robotics, our work has a broader impact on the general audience:

(i) The proposed method falls into the more general field of mechanical computing, which has attracted general audiences as indicated in the perspective article [3] recently published in *Nature*. Specifically, readers of *Nature Communications* have shown a strong interest in this area [4-6]. In accordance with reviewer 3, our contribution advances the field by providing a new way to enable computing units to integrate tightly with sensors and actuators, allowing the implementation of complete autonomous systems in a cut-and-fold process. We have added a discussion in the 2nd paragraph (lines 46-50) of the Introduction.

(ii) Our work also builds up a versatile robotic platform, which can further motivate multidisciplinary collaboration, spanning mechanics, electrical engineering,

material science, and manufacturing. For example, our method encourages the creation of novel artificial muscles that could be incorporated into our system to improve performance, e.g., energy efficiency and switching speed. Innovative fabrication methods and systematic modeling could also be motivated and involved to achieve high-throughput creation. We have included relevant discussions in the 3rd (line 446-464), 9th (line 528-537), and last paragraphs (line 559-564) of the Discussion.

(iii) In addition, thanks to the high accessibility of materials and ease of fabrication, our method has potential applications in the education and toy industry. These applications would be of great interest to a general audience. For instance, our folding-based robots can be used as teaching tools for resource-constrained students to learn through making, which is well aligned with the approach of “Maker education”. We added a discussion on this in the 7th paragraph (line 508-511) of the Discussion.

For the reviewer’s specific suggestions and comments, we have made corresponding revisions as below:

1. First, as the authors mentioned in the Introduction, there are several challenges in using rigid semiconductor-based electronics. For instance, semiconductor-based electronics are typically vulnerable to environmental factors, e.g., radiation and physical impact, which limit their applications. The authors should demonstrate that their origami robots can endure various environmental factors (radiation and physical impacts), proving their approaches with clear advantages over the origami robots with semiconductor-based electronics.

Response: Thanks for pointing this out. We used the flytrap-inspired robot as an example to demonstrate the advantages of our origami-based control over the semiconductor-based one. We operated the robots with two different controls under four environmental factors, i.e., static magnetic field, radio frequency signal, electrostatic discharge, and mechanical deformation. The results show that our origami-based robot behaved robustly as expected while the semiconductor-based one malfunctioned or even failed permanently. Although we do not have access to a radiation generator for the direct test, radiation damage to semiconductor-based devices, especially silicon-based ones, has been well identified and investigated [7], which further shows the advantage of our origami-based method in such environments. We have included the result in the 3rd paragraph (line 315-324) of Section: Flytrap-inspired prey-catching robot and added a new panel J in Fig. 5 (also see Supplementary Movie S9-12). More details about the robots and their comparison are also included in Supplementary Table S3 and Supplementary Text S9 (line 994-1036).

2. In the Introduction, the authors also mention that equipping external semiconductor-based electronics requires system integration that would increase the complexity and weight. The authors should make head-to-head comparisons of robotic weights and energy consumption between their origami robot systems and conventional origami robots with semiconductor-based electronics.

Response: Thanks for the helpful suggestion. As we mentioned in the last response above, we have used the flytrap-inspired robot to show the robustness of our origami-based method to environmental factors. We have also included a head-to-head

comparison of complexity (component count) and weight as described in Supplementary Table S3 and Supplementary Text S9 (line 994-1036). The results show that our method has obvious advantages in terms of weight.

We also compared the cost of two robots, although it is difficult to make an accurate comparison. Here we evaluate the total cost of two robots by calculating the corresponding off-the-shelf components as shown in Supplementary Table S3.

Additionally, the results demonstrate the benefits of using origami-based control. For more information, please refer to Supplementary Table S3 and Supplementary Text S9 (line 994-1036).

Although the energy consumption of the resulting robots is not the focus of this work, our CSCP actuators consume much more power compared to servo motors due to their low energy efficiency caused by the thermal-driven mechanism. This issue makes our systems suitable for applications where they are not constrained to power sources. We have added relevant discussions in the 7th paragraph (line 497-500) of the Discussion.

3. From Figure 1c–1e, most of the control units and electric power devices were installed on planar and non-actuatable locations. What are the advantages of the proposed cut-and-fold processing in comparison with the post-installation approach? I believe that conventional origami robots also install semiconductor-based electronics on planar and non-actuatable locations. Then, it became unclear to me the major advantages of the reported work.

Response: We thank the reviewer for bringing this up. Although the control units and electric power in our system are still installed at planar and non-actuatable locations, our method removes the reliance on rigid semiconductor-based components, which could lead to more robust origami robots. Our origami components are compliant, which enables the resulting robots to sustain large deformation and thus restore their functionalities afterward. On the contrary, mechanical bending and twisting could destroy rigid semiconductor-based electronics, resulting in the failure of robots. In addition, our method allows the creation of semiconductor-free origami robots, which are robust to adversarial environmental events, including static magnetic field, RF signal, and ESD. Although not the focus of this work, our work also leads to the future potential to create autonomous robots purely through folding from patterned 2D precursors without requiring post-installation. To achieve this goal, we need to develop versatile 2D composites that can tightly integrate the necessary functional materials to compose essential functionalities, including sensing, computing, and actuation. We have added relevant discussions in the 7th (line 497-511) and 9th paragraphs (line 528-537) in Discussion and Supplementary Text S1 (line 840-842) and S9 (line 994-1036).

4. Besides the artificial muscle actuators, can the authors' origami robot adopt other actuation systems, such as pneumatic systems or shape memory alloys?

Response: Thanks for asking. Yes, we can use other actuators to implement our systems. To enable easy cascading, we choose to use the same type of signal for both inputs and outputs of OMS-based devices. This can guarantee that the output signals

from the previous gate can be directly fed as the input of the next gate. Thus, the main requirements for actuators in our system are two: 1) they can be activated by the signals of interest. For example, our CSCP actuators can be electrically driven, and 2) actuators can generate enough driving displacement and force to toggle the bistable mechanisms. Taking SMA actuators as an example, they can be electrically operated and generate large linear contracting strokes, which makes them suitable for our systems. Other electrically driven actuators, e.g., conductive liquid crystal elastomers, are also applicable. Linear contracting pneumatic actuators are feasible to activate the bistable beam as shown in Ref. [8]. As a result, we need to use pneumatic current as a signal instead of electricity. Otherwise, we could build hybrid origami logic systems for specific applications that require different input and output signals, although additional interfacing components are required for cascaded circuits. For example, we can create origami logic with pneumatic inputs but electrical outputs similar to Ref.[9]. Alternative actuators, e.g., light-driven LCEs, may be adopted into our system to build more hybrid origami logic and autonomous machines. We have included relevant discussions in the 3rd paragraph (line 446-464) of the Discussion.

5. The authors should provide a table to summarize the performance and properties of origami robots and soft robots in the literature, and the readers can clearly understand the technological advances of this work.

Response: We appreciate the reviewer's valuable suggestions. We have added Supplementary Table S4 in Supplementary Materials to summarize the performance

and properties of autonomous origami robots and soft robots to show these advantages of our work.

Reviewer #3 (Remarks to the Author):

This paper describes a mechanical computing device that exploits buckling to achieve bistable behaviors and act as a switch. The so-called Origami Multiplexed Switch (OMS) can be manufactured by cutting and folding of paper, self-adhesive copper strips, and conductive thread.

The bistable beam can be actuated by two "fishing line" actuators and switches input current between two outputs, depending on its position. The authors show how these building blocks can be configured to implement basic logic gates, including NAND, which enables Turing universal computation.

While the OMS falls into a larger class of mechanical computing devices, this contribution is particularly interesting as the OMS integrates well with sensors and actuators, thereby allowing the realization of complete autonomous systems. This is demonstrated with three origami robots: a venus flytrap that requires two distinct physical contacts, a vehicle that reverses direction upon contact, and a second vehicle whose trajectory is programmed by a series of set-reset memory devices. Here, physical forces onto the robot translate into activation of a bistable beam and hence trigger the programmed behaviors. Although the demonstrated robots are very limited, the authors make a compelling case that other "smart polymers" can be used for sensing and activating the bistable beams and/or replacing the fishing line actuators

altogether. As such, the OMS concept might find a variety of applications in creating stimuli-responsive materials that go beyond purely reactive behavior but can exploit memory and computation.

Response: Thanks for confirming our contribution.

Supplementary materials describe the manufacturing process, a detailed analysis of the OMS dynamics (with switching times in the order of seconds), and an exploration of the possible design space that OMS can enable together with other means of sensing and actuation. Methods used are described in sufficient detail, albeit relaying on previous publications of the authors for some aspects of the manufacturing process.

Response: We appreciate the reviewer's acknowledgment of our contribution and support for publishing in Nature Communications.

Reference

[1] Yan, Wenzhong, and Ankur Mehta. "A cut-and-fold self-sustained compliant oscillator for autonomous actuation of origami-inspired robots." *Soft Robotics* 9, no. 5 (2022): 871-881.

[2] Burri, Jan T., Eashan Saikia, Nino F. Läubli, Hannes Vogler, Falk K. Wittel, Markus Rüggeberg, Hans J. Herrmann, Ingo Burgert, Bradley J. Nelson, and Ueli Grossniklaus. "A single touch can provide sufficient mechanical stimulation to trigger Venus flytrap closure." *PLoS biology* 18, no. 7 (2020): e3000740.

[3] H. Yasuda, P. R. Buskohl, A. Gillman, T. D. Murphey, S. Stepney, R. A. Vaia, and J. R. 683 Raney, Mechanical computing, *Nature* 598, 39–48 (2021)

- [4] El Helou, Charles, Philip R. Buskohl, Christopher E. Tabor, and Ryan L. Harne. "Digital logic gates in soft, conductive mechanical metamaterials." *Nature communications* 12, no. 1 (2021): 1-8.
- [5] Song, Yuanping, Robert M. Panas, Samira Chizari, Lucas A. Shaw, Julie A. Jackson, Jonathan B. Hopkins, and Andrew J. Pascall. "Additively manufacturable micro-mechanical logic gates." *Nature communications* 10, no. 1 (2019): 1-6.
- [6] Mahboob, I., E. Flurin, K. Nishiguchi, A. Fujiwara, and H. Yamaguchi. "Interconnect-free parallel logic circuits in a single mechanical resonator." *Nature communications* 2, no. 1 (2011): 1-7.
- [7] Spieler, Helmuth. "Introduction to radiation-resistant semiconductor devices and circuits." In *AIP Conference Proceedings*, vol. 390, no. 1, pp. 23-49. American Institute of Physics, 1997.
- [8] Yang, Dian, Mohit S. Verma, Ju-Hee So, Bobak Mosadegh, Christoph Keplinger, Benjamin Lee, Fatemeh Khashai, Elton Lossner, Zhigang Suo, and George M. Whitesides. "Buckling pneumatic linear actuators inspired by muscle." *Advanced Materials Technologies* 1, no. 3 (2016): 1600055.
- [9] Garrad, M., Gabor Soter, A. T. Conn, Helmut Hauser, and Jonathan Rossiter. "A soft matter computer for soft robots." *Science Robotics* 4, no. 33 (2019): eaaw6060.

REVIEWERS' COMMENTS

Reviewer #1 (Remarks to the Author):

I am satisfied that the authors have addressed my concerns from the first round of reviews.

Reviewer #2 (Remarks to the Author):

The authors have revised the manuscript based on the reviewers' comments and suggestions, and I am now satisfied with the current version to be published in Nature Communications.

Response letter for NCOMMS-22-36157A

REVIEWER COMMENTS

REVIEWERS' COMMENTS

Reviewer #1 (Remarks to the Author):

I am satisfied that the authors have addressed my concerns from the first round of reviews.

Response: We are grateful to hear that the reviewer is satisfied with our revised work, and we appreciate the acceptance of our submission.

Reviewer #2 (Remarks to the Author):

The authors have revised the manuscript based on the reviewers' comments and suggestions, and I am now satisfied with the current version to be published in Nature Communications.

Response: Thank you for agreeing to publish our work in Nature Communications.